# Effects of Diet on the Gut Bacterial Community of *Aldrichina grahami* (Diptera: Calliphoridae) across Developmental Stages

**DOI:** 10.3390/insects15030181

**Published:** 2024-03-07

**Authors:** Zhen Li, Chao Yue, Na Ma, Guanjie Yan

**Affiliations:** 1Henan International Joint Laboratory of Insect Biology, Nanyang Normal University, Nanyang 473061, China; lizhenmail1118@163.com (Z.L.); yuechaomail@163.com (C.Y.); 2Henan Key Laboratory of Insect Biology in Funiu Mountain, Nanyang Normal University, Nanyang 473061, China; 3College of Life Science and Agricultural Engineering, Nanyang Normal University, Nanyang 473061, China; 4Henan Field Observation and Research Station of Headwork Wetland Ecosystem of the Central Route of South- to-North Water Diversion Project, Nanyang Normal University, Nanyang 473061, China

**Keywords:** Firmicutes, Proteobacteria, functional predictions, blowfly

## Abstract

**Simple Summary:**

The blowfly, *Aldrichina grahami*, is widely recognized as a highly significant insect in forensic investigations. Despite the pivotal role of gut microbes in various facets of insect biology, limited knowledge exists regarding the gut microbiome of *A. grahami*. To investigate the gut bacterial community of *A. grahami* and explore its stability, diet and developmental stage were selected as the two variables. Larvae were reared on bovine liver, swine manure, and chicken manure, and high-throughput sequencing was performed on the 1st, 2nd, and 3rd instar larvae and the newly emerged adults. Our results revealed that the dominant genera *Vagococcus*, *Providencia*, *Lactobacillus*, and *Morganella* were present in more than 80% of gut samples. The alpha and beta diversity of the gut bacterial community do not vary significantly across different diets and developmental stages. Our results indicated that the bacterial community in the gut of *A. grahami* exhibits remarkable stability, and the dominant genera *Vagococcus*, *Providencia*, *Lactobacillus*, and *Morganella* might be potential core microbes in the gut microbial community.

**Abstract:**

The blowfly, *Aldrichina grahami* (Diptera: Calliphoridae), is a well-known forensically important insect. Basic data related to *A. grahami* have been well documented; but despite the pivotal role of gut microbes in various facets of insect biology, little is known about its gut microbiome. To investigate the gut bacterial community of *A. grahami* and explore its stability, diet and developmental stage were selected as the two variables. Larvae were reared on bovine liver, swine manure, and chicken manure, and high-throughput sequencing of the 1st, 2nd, and 3rd instar larvae and the newly emerged adults was performed. According to our results, the alpha diversity of the gut bacterial community did not significantly vary among different diets and developmental stages. Principal coordinate analysis revealed that the gut microbiome of *A. grahami* clustered together among different diets and developmental stages. The main phyla in the gut microbial community of *A. grahami* were Firmicutes and Proteobacteria, and the dominant genera were *Vagococcus*, *Providencia*, *Lactobacillus*, and *Morganella*. These findings characterized the gut microbiome of *A. grahami* and demonstrated that the gut bacterial community is fairly stable. The dominant genera *Vagococcus*, *Providencia*, *Lactobacillus*, and *Morganella* have the potential to serve as core microbiomes within the gut microbial community of *A. grahami*.

## 1. Introduction

The blowfly, *Aldrichina grahami* (Diptera: Calliphoridae), is a well-known forensically important insect that commonly forms the first wave of sarcosaphagous flies to arrive and oviposit into animal carcasses in early spring and late autumn [1]. Basic data related to *A. grahami*, such as information about the effects of temperature on its development [2,3], its pupal morphological changes [4], and the effects of methamphetamine (MA) on its development [5], have been well documented; but few studies have been conducted on its gut microbial community. Symbiotic microorganisms in the insect gut are related to many host aspects, such as metabolism, degradation of harmful substances [6], nutritional provision [7,8], protection against pathogens [9,10], intra- and interspecies communication [11,12], mating and reproductive capabilities [13], immune formation, and maturity [14]. Therefore, it is necessary to investigate the gut microbial community of *A. grahami* to establish a comprehensive framework, given the utility of this insect for forensic or nutritional purposes.

The gut microbial community of insects shows both stability and plasticity. On the one hand, to ensure essential catabolic abilities for health and survival, a common microbial core community can be identified within some insect species [15]. For example, the gut microbiome of *Periplaneta americana* (Blattaria: Blattidae) exhibits a remarkably stable core microbiome with minimal structural alterations and low variance in response to dietary changes [16]. On the other hand, certain insect species have been shown to lack indigenous gut microbiota [17], and the composition of the gut bacterial community can also be influenced by intrinsic factors of the host, including age, sex, genotype, and exogenous factors [18,19]. Thus, one objective of this study was to investigate the stability of the gut bacterial community in *A. grahami*.

Diet represents one of the exogenous factors capable of modulating host–microbial interactions, thereby influencing the composition and functional associations within animal gut microbial communities [20,21]. Diverse diets can exhibit variations in their macronutrient composition, thereby potentially promoting specific bacterial communities within the host. For example, studies of the housefly and red palm weevil have shown that gut microbial communities vary across different diets [22,23]. Moreover, diet-associated microbes may also represent a source of potential colonizers of the gut that are transmitted vertically from parent to offspring [24].

For holometabolous insects, all the larval organs, including those of the gut, are broken down and reconstructed [25,26,27,28]. In addition, the larval and imago stages have different feeding activities and gut environments, which can influence their gut bacterial symbionts [29]. Thus, in addition to diet, developmental stage also dramatically affects the host’s gut microbes. For example, Firmicutes were the main bacterial community in the 3rd and 5th larval instars of *Spodoptera exigua* (Lepidoptera: Noctuidae), while Proteobacteria dominated the other developmental stages [30]. Culturable bacteria exhibited variations across different developmental stages of *Plutella xylostella* (Lepidoptera: Plutellidae), with a total of twenty-five distinct bacterial strains identified, including 20 strains in the larval gut, 8 strains in pupae, and 14 strains in emerged adults [31]. Therefore, diet and developmental stage were chosen as the two variables for assessing the stability of the bacterial community in *A. grahami*.

In this study, we reared *A. grahami* larvae with bovine liver (BL), swine manure (SM), and chicken manure (CM) and investigated the microbial community composition of *A. grahami* in different developmental stages (1st–3rd stages and newly emerged adults) by high-throughput amplicon sequencing of a variable region of the bacterial 16S rRNA gene, in order to test (1) the bacterial community composition of the *A. grahami* gut, and (2) the stability of the gut bacterial community across different diets and developmental stages.

## 2. Materials and Methods

### 2.1. Fly Stock and Rearing

In this study, flies were obtained from three fly traps baited with bovine liver for next-generation breeding in Nanyang, China, in May 2022. Sixty *A. grahami* were transferred to a cage (35 × 35 × 35 cm) and reared under 12:12 (L:D) h photoperiods at 30–40% relative humidity and 24–26 °C room temperature, after which water and blood were extracted from the bovine liver (offered from 9 a.m. to 10 a.m. daily). For egg collection, three oviposition sites were established by adding approximately 20 g of bovine liver (BL), swine manure (SM), or chicken manure (CM) to a plastic tray (15 × 15 cm). The resulting eggs were subsequently used for F1 generation rearing, and three cages were established with each resource.

For larval rearing of BL groups, about 300 eggs collected from bovine liver were transferred to another bovine liver (about 50 g) in a plastic container (9 cm diameter × 7 cm height) that was placed on dry sand (about 5 cm deep) in a plastic box (30 × 25 × 15 cm). The box was enclosed with mesh to prevent the larvae from escaping. The plastic box was maintained in a controlled incubator (25 ± 0.5 °C) and bovine liver (100–200 g) was introduced daily at 9 a.m. until all larvae had reached the wandering stage (they climbed out of their plastic boxes and searched for a place to pupate). When all larvae had pupated, they were incubated for further 4 days. Then, pupae were moved to a plastic container and covered with about 2 cm of dry sand. The plastic container was placed in a 35 × 35 × 35 cm cage, which was maintained under a photoperiod of 12:12 (L:D) h at a room temperature of 24–26 °C and relative humidity of 30–40%. The larval rearing of SM and CM groups was performed with similar operations, only replacing the bovine liver with swine manure or chicken manure. The larvae utilized for gut sample collection were randomly selected from each group and subjected to a 12 h starvation period in sterile centrifuge tubes to deplete their gut contents.

### 2.2. Sample Collection

The newly emerged adults and larvae were paralyzed at 4 °C and moved to a sterile environment for sample collection. Adults and larvae were subjected to surface sterilization by alternating washing with water and 75% ethanol for three cycles, then third instar larvae and adults were dissected in a sterile environment to sample their guts. First and second instar larvae, or the intestines of third instar larvae and adults, were rapidly moved to 10 mL sterile EP tubes. The three diet products—fresh bovine liver, chicken manure, and swine manure—were also transferred to 10 mL sterile EP tubes. Each treatment had three replicates, and all the samples were stored in a −80 °C freezer until further testing. Limited by body size, 30–50 whole individuals of first and second instar larvae were used as replicates. For third instar larvae and newly emerged adults, the guts of 20 individuals were dissected and pooled in a centrifuge tube as a replicate.

### 2.3. DNA Extraction and Sequencing Library Construction

According to the manufacturer’s instructions, the E.Z.N.A.^®^ soil DNA kit (Omega Bio-tek, Norcross, GA, USA) was used for microbial DNA extraction. The final determination of DNA concentration and purity was performed using a NanoDrop 2000 spectrophotometer (Thermo Fisher Scientific, Waltham, MA, USA), while the DNA quality check was conducted through 1% agarose gel electrophoresis. Subsequently, the V3–V4 variable region of the 16S rRNA gene was amplified by PCR utilizing the aforementioned extracted DNA as a template. The primers used for amplification were 338F (5′-ACTCCTACGGGAGGCAGCAG-3′) and 806R (5′-GGACTACHVGGGTWTCTAAT-3′), which carried a Barcode sequence.

The PCR was performed in triplicate using a 20 μL reaction mixture consisting of 5 × TransStart FastPfu Buffer (4 μL), 2.5 mM dNTPs (2 μL), an upstream primer (5 μM, 0.8 μL), a downstream primer (5 μM, 0.8 μL), TransStart FastPfu Polymerase (0.4 μL), and template DNA (10 ng). The amplification procedure commenced with pre-denaturation at 95 °C for 3 min, followed by denaturation at 95 °C for 30 s during each of the subsequent 27 cycles, annealing at 55 °C for 30 s, and extension at 72 °C for 30 s. Subsequently, a stable extension step was carried out at 72 °C for an additional duration of 10 min. Finally, the samples were stored at a temperature of 4 °C (PCR instrument used: ABI GeneAmp^®^ 9700, AppliedBiosystems, Foster City, CA, USA). The experimental design included three replicates for each sample.

The PCR products obtained from the same sample were consolidated and retrieved using a 2% agarose gel, followed by purification of the retrieved products utilizing the AxyPrep DNA Gel Extraction Kit (Axygen Biosciences, Union City, CA, USA). Following purification, the retrieved products were subjected to analysis through 2% agarose gel electrophoresis and quantification using the Quantus™ Fluorometer (Promega Corporation, Madison, WI, USA).

A library consisting of purified PCR products was constructed using the NEXTFLEX Rapid DNA-Seq Kit (PerkinElmer, Austin, TX, USA). The construction process involved four steps: (1) ligation of adapters; (2) magnetic bead-based selection to eliminate chimeric artifacts; (3) PCR amplification for template enrichment; and (4) final library retrieval, achieved through magnetic bead purification. Sequencing was conducted on Illumina’s Miseq PE300/NovaSeq PE250 platform (Shanghai Meiji Biomedical Technology Co., Ltd., Shanghai, China). The raw data have been deposited in the NCBI Sequence Read Archive (SRA) database (accession numbers SRR26831487-SRR26831522).

### 2.4. High-Throughput Sequencing Data Analysis

The raw fastq [32] files underwent quality filtering using Trimmomatic and were subsequently merged using FLASH (version 1.2.11) [33] based on the following criteria: (i) reads with an average quality score <20 over a 50 bp sliding window were truncated; (ii) sequences with an overlap longer than 10 bp were merged if their overlap had no more than 2 bp mismatches; and (iii) sequences from each sample were separated based on barcodes that required exact matches, while allowing for up to 2 nucleotide mismatches in primers. Reads containing ambiguous bases were excluded.

The operational taxonomic units (OTUs) were clustered using UPARSE [34] with a novel ‘greedy’ algorithm, which simultaneously performs chimera filtering and OTU clustering at a 97% similarity cutoff. Taxonomic annotation of the OTU species was conducted by employing the Ribosomal Database Project (RDP) classifier in comparison with the Silva 16S rRNA gene database (v138), utilizing a confidence threshold of 70%. The community composition of each sample was determined at various levels of species classification. For 16S function prediction analysis, PICRUSt2 software (version 2.2.0) was employed.

### 2.5. Alpha Diversity and Beta Diversity

All data analysis was conducted on the biological cloud platform (https://cloud.majorbio.com, accessed on 2 March 2023) and was performed as described below: the mothur software (http://www.mothur.org/wiki/Calculators, accessed on 2 March 2023) was utilized to compute Alpha diversity metrics (Chao 1 and Shannon index). The Wilcoxon rank-sum test was employed for assessing the differences in Alpha diversity between groups. Based on the Bray–Curtis distance algorithm, principal coordinate analysis (PCoA) was conducted to evaluate the similarity of microbial community structures among samples, while the PERMANOVA non-parametric test was applied to determine the significance of differences in microbial community structures between samples. Additionally, LEfSe analysis (Linear discriminant analysis Effect Size) (http://huttenhower.sph.harvard.edu/LEfSe, accessed on 2 March 2023), with LDA > 2 and *p* < 0.05 thresholds, was performed to identify bacterial taxa exhibiting significant abundance variations from phylum to genus level across different groups.

## 3. Results

### 3.1. Sequencing Data

A total of 2,188,791 high-quality sequences were obtained from 45 extracted DNA samples, extracted from the three diets and from the gut using amplicon sequencing targeting the 16S rRNA gene. After quality and abundance filtering, an average of 48,640 reads per sample were retained, with an average sequence length of 424 bp (Appendix A).

These results demonstrate the suitability of our findings for analyzing the microbial community compositions of diverse insect gut samples. Additionally, the rarefaction curve revealed variations in OTU richness among all gut samples, with patterns approaching saturation (Figure 1). The high coverage indices of all the *A. grahami* samples using Good’s method indicated that the sequencing depth was sufficient for profiling the bacterial communities present (Appendix A).

### 3.2. Species Richness and Diversity

The gut bacterial community was identified at different taxonomic levels. Phyla with sporadic occurrence and low abundance in certain samples were categorized as ‘others’ (abundance < 1%). The gut microbial community of *A. grahami* was dominated by the phylum Firmicutes (61.3% relative abundance), followed by Proteobacteria (32.2%), Bacteroidota (4.0%), and Actinobacteriota (2.4%; Figure 2a). Within the phylum Firmicutes, organisms belonging to the *Vagococcus* and *Lactobacillus* genera were dominant (Figure 2b). The majority of the Proteobacteria belonged to the genera *Providencia* and *Morganella* (Figure 2b).

### 3.3. Comparison of Bacteral Communities

The variations in the bacterial communities were reflected by the alpha diversities (Figure 3a,b). The Chao1 and Shannon indices of chicken manure and the Chao1 index of swine manure were significantly greater than those of 2nd instar larvae and adults in the BL group, and the Shannon index of swine manure was significantly greater than that of adults in the BL group. However, neither the Chao 1 indices nor the Shannon indices were significantly different among the developmental stages or diet treatments (Figure 3a,b). In addition, the PCoA analysis revealed that the bacterial communities of the gut were not independent of each other across the different diets and developmental stages, but the bacterial communities of the three diets (bovine liver, chicken manure, and swine manure) were independent of each other and their related gut samples (Figure 3c; Appendix A). Moreover, the Venn diagram at the genus level showed that 16 genera coexisted in all gut samples (Figure 3d), including the genera *Vagococcus*, *Providencia*, *Lactobacillus*, and *Morganella*, the top 4 genera in terms of relative abundance (Figure 2b; Appendix A).

**Figure 2 insects-15-00181-f002:**
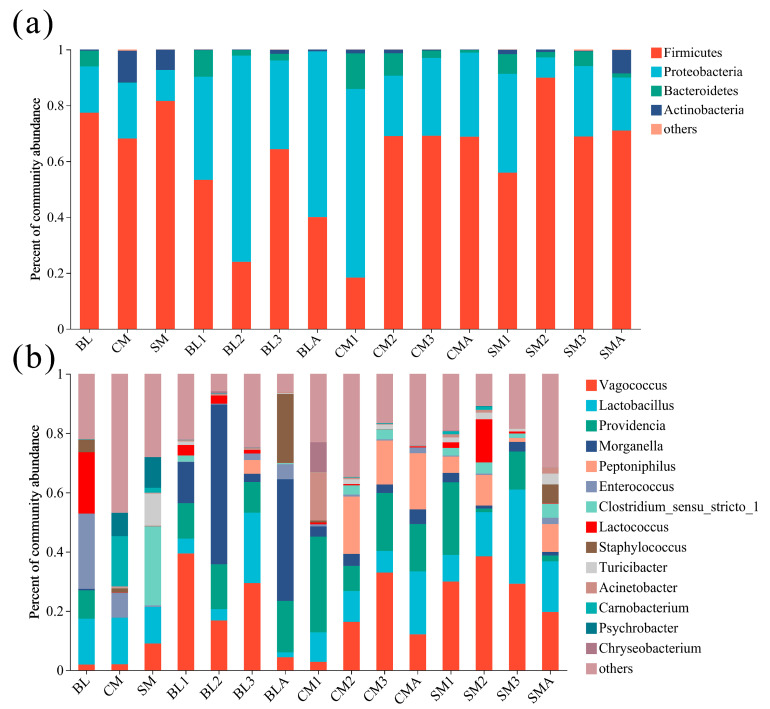
Relative abundance of bacteria composition at (**a**) phylum, and (**b**) genus level. BL: bovine liver; CM: chicken manure; SM: swine manure; BL (1–3): 1–3 instar larvae fed with bovine liver; CM (1–3): 1–3 instar larvae fed with chicken manure; SM (1–3): 1–3 instar larvae fed with swine manure; BLA, CMA, and SMA: newly emerged adults developed on bovine liver, chicken manure, and swine manure diets. Taxa with an abundance of <1% for phylum and 5% for genus are included in “others”.

According to the LEfSe analysis (Figure 4a), most of the gut microbial communities of *A. grahami* were not significantly affected by different diets or developmental stages. The relative abundances of the *Proteus* genus in the BL diet; the Bacteroidetes phylum, Flavobacteriia class, Flavobacteriales order, and Flavobacteriaceae family in the CM diet; and the *Empedobacter* genus in the SM diet were significantly enriched in the 1st instar larvae (Figure 4a,b).

### 3.4. Functional Predictions of the Gut Microbiome of A. grahami

The abundances of KEGG pathways were predicted using PICRUSt2 based on 16S rRNA gene sequences, and the pathway abundances were visualized through heatmaps at two hierarchical levels (Figure 5). At pathway level 1, metabolism accounted for approximately 72.85–75.19% of the metabolism in each group (Figure 5a). At level 2, the richness of the top 20 pathways is shown. The relative abundance of carbohydrate metabolism (8.24–10.86%) was the highest among the different groups, followed by amino acid metabolism (5.87–7.11%; Figure 5b).

## 4. Discussion

In this study, the dynamics of the gut microbiome of *A. grahami* were measured. The gut bacterial communities in *A. grahami* were dominated by the phyla Firmicutes and Proteobacteria. Moreover, organisms of the phylum Firmicutes belong to the *Vagococcus* and *Lactobacillus* genera, while the majority of Proteobacteria belong to the *Providencia* and *Morganella* genera (Figure 2b). Similarly, Junqueira et al. [35] reported that the phyla Proteobacteria, Bacteroidetes, and Firmicutes were the predominant organisms in the microbiomes of both blowflies and houseflies. A study on the bacterial community structures associated with different stages of other blowflies, *Lucilia sericata* and *Lucilia cuprina,* also showed that, for colony-shaped flies reared on beef liver, the majority of the bacteria were from the phyla Proteobacteria and Firmicutes [36]. *Providencia*, *Lactobacillus*, and *Morganella* are the most common genera in the gut microbiomes of blowflies and houseflies [35,36,37]. *Vagococcus* was reported as one of the most dominant genera in the gut of *Phormia regina* (Diptera: Calliphoridae) [38]. These results indicate that the four dominant genera—*Vagococcus*, *Lactobacillus*, *Providencia*, and *Morganella*, which were identified in the guts of *A. grahami*—are commonly found in other fly species.

Our results showed that the alpha diversity of the gut bacterial community of *A. grahami* was not significantly affected by diet or developmental stage (Figure 3a,b). In addition, the PCoA analysis revealed that the bacterial communities of the gut samples were independent across different diets and developmental stages (Figure 3c). Furthermore, the PCoA analysis revealed that the gut bacterial communities obtained from each dietary group exhibited similar characteristics, independent of the diet (Appendix A). Our results indicated a highly stable microbial community of *A. grahami* across different diets and developmental stages. Similarly, the gut bacterial communities of the soldier fly (*Hermetia illucens*) and cockroach (*Periplaneta americana*) showed highly stable core microbial communities, with low variance in composition in response to dietary shifts [16,39], and a study of houseflies also showed that the richness and structure of the gut bacterial community did not vary significantly among different developmental stages [40]. However, the gut bacterial communities of the gypsy moth (*Lymantria dispar*), cotton bollworm *(Helicoverpa armigera*), fruit fly (*Drosophila melanogaster*)*,* and plant bug (*Adelphocoris suturalis*) have been extensively studied to reveal significant diet-related effects [41,42,43,44,45]. Additionally, developmental stage-dependent variations have been well documented for the gut bacterial community in fruit borers (*G. molesta*), melon flies (*Zeugodacus cucurbitae*), plant bugs (*A. suturalis*)*,* and leaf beetles (*Gastrolina depressa*) [46,47,48,49]. Interestingly, insects such as gypsy moths, fruit flies, and beetles, which exhibit varied gut bacterial communities across different diets or development stages, typically have relatively simple diets and inhabit environments with simple bacterial compositions. In contrast, insects such as *A. grahami*, the soldier fly, and the cockroach, which exhibit stable gut bacterial communities across different diets or development stages, typically have mixed diets and inhabit environments with complex bacterial compositions. These findings suggest that insects inhabiting low-bacteria environments tend to exhibit a heightened susceptibility of their gut bacterial communities to environmental influences, whereas insects residing in complex bacterial environments tend to exhibit a more stable composition of their gut microbiomes.

A microbial core community can commonly be identified within a taxonomic entity to ensure essential catabolic aptitudes for health and survival [15]. The genera *Actinomyces*, *Dysgonomonas*, and *Enterococcus* were assumed to constitute a core community of the black soldier fly (*H. illucens*) based on their presence in at least 80% of the gut samples, with no significant impact from different diets observed on the gut microbiome [39]. Our results showed that the top four genera in terms of the relative abundances of *Vagococcus*, *Providencia*, *Lactobacillus*, and *Morganella* were present in at least 80% of the gut samples, and the relative abundances of these four genera and the two phyla did not significantly vary across the different diets and developmental stages of *A. grahami* (Appendix A). Moreover, all the gut samples contained sixteen genera, with one absent in the bovine liver diet, two absent in the chicken manure diet, and four absent in the swine manure diet (Figure 3d, Appendix A), which implies that larvae possess the capacity to selectively modulate specific gut genera. Overall, these results indicate that the genera *Vagococcus*, *Providencia*, *Lactobacillus*, and *Morganella* may constitute potential core members of the gut microbial community in *A. grahami*, and larvae exhibit the capacity to selectively screen for specific gut microbiota.

Previous studies on gut samples from diverse insect species have demonstrated that the phyla Firmicutes and Proteobacteria are involved in the degradation of animal manure and play important roles in degrading complex plant carbohydrates [39,45,46]. *Vagococcus* and *Providencia* have been reported to be more abundant in black soldier fly vermicomposting than in natural composting and could have strong metabolic abilities [50]. The functional predictions of the microbiome in the gut of *A. grahami* in this study also showed that metabolism accounted for approximately 72.85%–75.19% of the bacteria in each group (Figure 5a), and the relative abundance of carbohydrate metabolism (8.24%–10.86%) was the highest among the different groups, followed by amino acid metabolism (5.85%–7.10%; Figure 5b). These results indicate that the gut microbial community of *A. grahami* is strongly metabolically related.

In summary, we investigated the gut bacterial community of *A. grahami* and compared the variation in the gut microbiome of *A. grahami* across different diets and developmental stages. The alpha and beta diversity of the bacterial community was fairly stable among the different diets and developmental stages. The genera *Vagococcus*, *Providencia*, *Lactobacillus*, and *Morganella* might be potential core microbiomes in the gut microbial community of *A. grahami*, and the function of the microbiome in the gut of *A. grahami* is strongly metabolic-related. However, as all flies used in this study were obtained from the same location, identification of the core microbiome of *A. grahami* should be addressed in the future. The investigation of the gut bacterial community in *A. grahami* not only contributes to the enhancement of waste food conversion, but also facilitates the optimization and utilization of insects.

## Figures and Tables

**Figure 1 insects-15-00181-f001:**
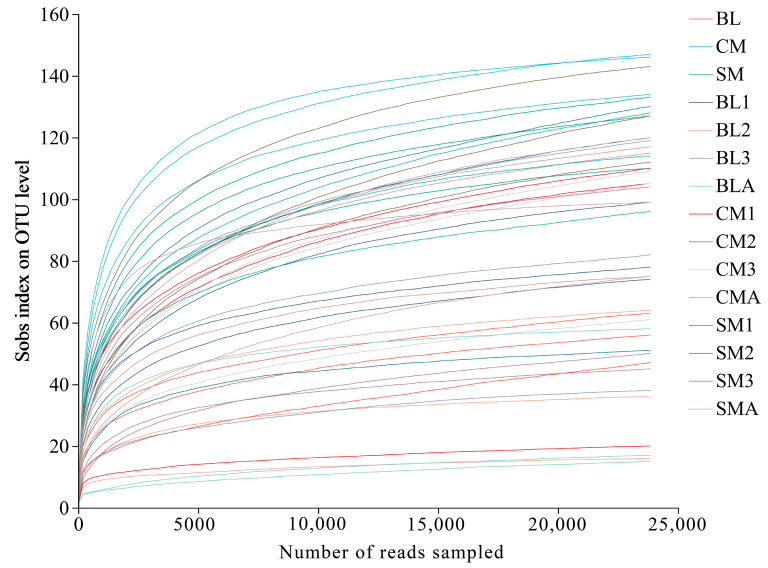
Sobs index rarefaction curves.

**Figure 3 insects-15-00181-f003:**
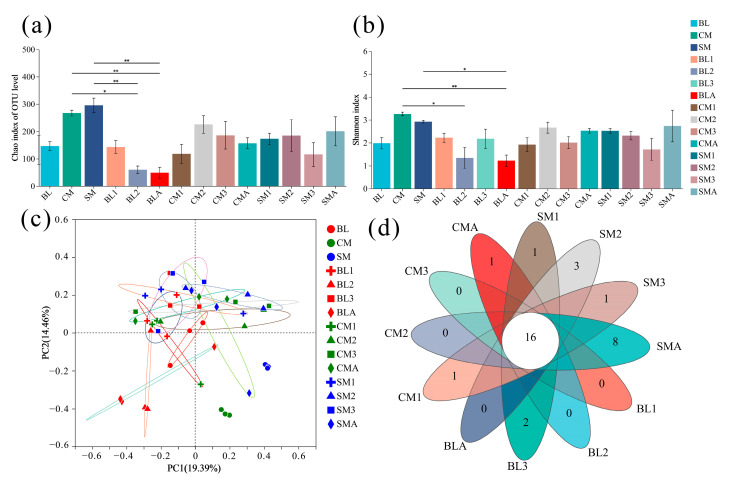
Effects of diet and developmental stage on the gut microbiome of *A. grahami*. (**a**) Chao 1, the error line represents the SD value, 0.01 < *p* < 0.05 was marked as *, *p* < 0.01 was marked as **, (**b**) Shannon values, the error line represents the SD value, 0.01 < *p* < 0.05 was marked as *, *p* < 0.01 was marked as **, (**c**) Principal coordinates analysis (PCoA) based on Bray–Curtis distance, and (**d**) Venn diagrams of gut microbiomes of *A. grahami* during different treatments. BL: bovine liver; CM: chicken manure; SM: swine manure; BL (1–3): 1–3 instar larvae fed with bovine liver; CM (1–3): 1–3 instar larvae fed with chicken manure; SM (1–3): 1–3 instar larvae fed with swine manure; BLA, CMA, and SMA: newly emerged adults developed on bovine liver, chicken manure, and swine manure diets.

**Figure 4 insects-15-00181-f004:**
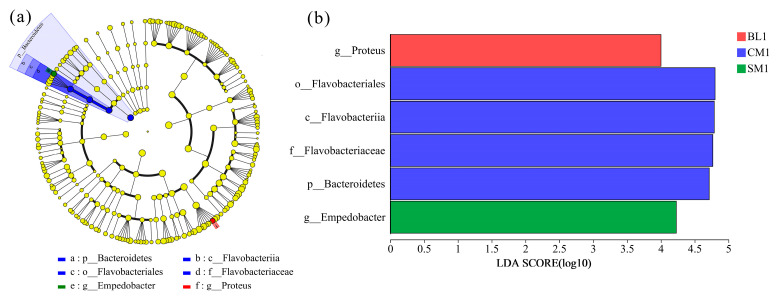
LEfSe analysis showing significant differences in gut microbial communities from phylum to genus. (**a**) Cladogram showing the phylogenetic distribution of gut microbial communities across different diets and life stages. Yellow nodes represent microbial taxa with no significant difference between different life stages, while other color nodes represent microbial taxa that are significantly enriched at those life stages. Regions in different colors represent different constituents. Circles indicate phylogenetic levels from phylum to genus. The diameter of each circle is proportional to the abundance of the group. (**b**) Bar diagram for different diets and life stages with LDA scores higher than 2.0.

**Figure 5 insects-15-00181-f005:**
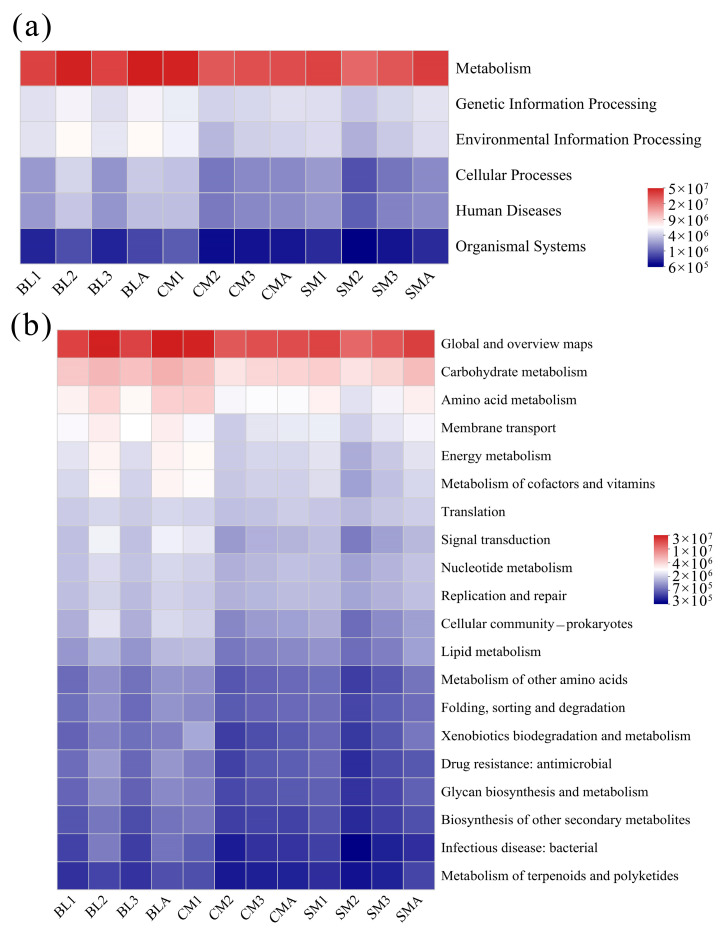
Prediction of KEGG functions of gut bacteria during different developmental stages in *A. grahami*. (**a**) Function prediction in pathway level 1, and (**b**) Function prediction, in pathway level 2.

## Data Availability

Raw data for the 16S rRNA gene have been deposited in the NCBI Sequence Read Archive (SRA) database (https://www.ncbi.nlm.nih.gov/ (accessed on 15 November 2023 and 5 March 2024), accession numbers SRR26831487-SRR26831522 and SRR28242891-SRR28242899). The data supporting the results may be found either in the manuscript or in the Appendix A.

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
