# Peer review of "Effects of Diet on the Gut Bacterial Community of Aldrichina grahami (Diptera: Calliphoridae) across Developmental Stages"

_insects, 2024, doi:10.3390/insects15030181_

Round 1

Reviewer 1 Report

Comments and Suggestions for Authors

The authors herein present the summary of the characterized gut microbiome of A. grahami.The submission is standardly written however, it is unclear from tge introduction and the discussion why this was done, or needed. No context was given for the study nor any rationale. This is evident in the way the results were presented and the subsequent  discussion.Overall, it is ,my recommendation that the submission undergoes extensive/major revisions before even consideration for acceptance

Specific thoughts are below:

 Line 17-20: Scientific name need to be italicized. This is basic and standard science practice.

Line 25: what does " host the first wave" mean?

Line 29: Define METH at the first instance of mention!!!!

Lines 33-35: This statement is wrong and written in a potentially misleading way. As is, it  implies all of those  functions are characterized in A. grahami, which then begs the question, then why bother do the work in this species? Please rephrase and use the citations correctly in this section and throughout the submission.

Line 36-40: This sentence and section does not make sense. Please rephrase to make your intent clearer.

Overall the introduction is lacking. I do not get any sense as to why  the need to characterize the gut microbiome in this species in response to diets is important other than it was "there". Whats the rationale for using these various diets and why choose these in partiuclar? Do they relate in some way to the ecology of the flies? are they bigger on various diets or is this standard practice? Overall the rationale for the study is lost to me and i dont think its the job of the reader to infer and assign rationale.

METHODS:

Its unclear if larval samples were surface sterilized. This has implications for  the data generated and how it is interpreted. The rest of the methods section is standard.

RESULTS

Comparisons of the larval gut microbiomes in relation to the environment or diets from which they were contained would have been informative as it demonstrates  selective screening of environmental microbes by the gut. The absence of this diet data reflects the absence of a context for this data. The rest of the results are standard. Yes, the plots are pretty but in the absence of a context or a need for this, it is essentially not clear what the value is.

DISCUSSION

This is lacking substance as there is no context  for the study and thus no context to discuss the results!!! what does this mean for the ecology of the larvae, /

Comments on the Quality of English Language

The authors herein present the summary of the characterized gut microbiome of A. grahami.The submission is standardly written however, it is unclear from tge introduction and the discussion why this was done, or needed. No context was given for the study nor any rationale. This is evident in the way the results were presented and the subsequent  discussion.Overall, it is ,my recommendation that the submission undergoes extensive/major revisions before even consideration for acceptance

Specific thoughts are below:

 Line 17-20: Scientific name need to be italicized. This is basic and standard science practice.

Line 25: what does " host the first wave" mean?

Line 29: Define METH at the first instance of mention!!!!

Lines 33-35: This statement is wrong and written in a potentially misleading way. As is, it  implies all of those  functions are characterized in A. grahami, which then begs the question, then why bother do the work in this species? Please rephrase and use the citations correctly in this section and throughout the submission.

Line 36-40: This sentence and section does not make sense. Please rephrase to make your intent clearer.

Overall the introduction is lacking. I do not get any sense as to why  the need to characterize the gut microbiome in this species in response to diets is important other than it was "there". Whats the rationale for using these various diets and why choose these in partiuclar? Do they relate in some way to the ecology of the flies? are they bigger on various diets or is this standard practice? Overall the rationale for the study is lost to me and i dont think its the job of the reader to infer and assign rationale.

METHODS:

Its unclear if larval samples were surface sterilized. This has implications for  the data generated and how it is interpreted. The rest of the methods section is standard.

RESULTS

Comparisons of the larval gut microbiomes in relation to the environment or diets from which they were contained would have been informative as it demonstrates  selective screening of environmental microbes by the gut. The absence of this diet data reflects the absence of a context for this data. The rest of the results are standard. Yes, the plots are pretty but in the absence of a context or a need for this, it is essentially not clear what the value is.

DISCUSSION

This is lacking substance as there is no context  for the study and thus no context to discuss the results!!! what does this mean for the ecology of the larvae, /

Author Response

Dear Editors and Reviewers:

Thank you for your letter and for the reviewers comments concerning our manuscript entitled “Effects of diet on the gut bacterial community of Aldrichina grahami (Diptera: Calliphoridae) across developmental stages” (ID: 2852905). We thank the reviewers for the time and effort that they have put into reviewing the previous version of the manu. Those comments are all valuable and very helpful for revising and improving our paper, as well as the important guiding significance to our researches. Based on the instructions provided in your letter, we uploaded the file of the revised manu. Appended to this letter is our point-by-point response to the comments raised by the reviewers. Revised portion are marked in red in the paper.We have studied comments carefully and have made correction which we hope meet with approval.

Once again, thank you very much for your comments and suggestions.

Reviewer 2 Report

Comments and Suggestions for Authors

The manuscript refers to investigations on the gut microbiome of larvae and adults of Aldrichina grahami, a forensically important insect that was reared on three different diets. Gut bacteria were analysed using the most advanced techniques, identified up to genus level and their diversity was measured in order to evaluate the effects of diet and developmental stage on the gut microbiome community. Overall, the work is well done and the results obtained from this research might be useful for all the scientists interested in studying relationships between insects and bacteria, mostly for specific metabolic activities carried out by these microorganisms.

The manuscript is well structured and balanced in their parts. In the introduction some citations are wrong or not relevant. I underlined some flaws and mistakes throughout the manuscript.

My only concern involves the discussion chapter. I would expect a deeper discussion when present results are compared to those obtained in previous research dealing with different insect species. I think this part should be improved.

Further comments and suggestions are detailed on a point to point basis in the attached file.

Author Response

(The authors gave the same response as above.)

Round 2

Reviewer 1 Report

Comments and Suggestions for Authors

The revised submission is signfcanlty more improved, but the introduction, results and discussion could use some further restructuring and a slght shift in focus given the data generated to make it more impressive and advance knoweldge.

1.    I appreciate the revisions made by the authors. The mansucript is much more coherent in this version. However, the introduction could use some further background. The mention of diet is a good adition, but the switch to manure has no hsitorical citations provided. Do we know if the different manusre result in different gut micprbiobiota in other insect or other sarcopjagous insects? If not, then framw the intro as the need to investigate this given the possible utility of this insect for forensic or nutritional purposes.

2.     I appreciate the inclusion of the diets in the results but fail to see the comparison (atleast in an NMDS plot and mayeb even an alpha diversity comparison) of the communities in the diet and how they differ from the insects. This would indicate some selective screening lending credenc to the presence of a “core stabel “ gut microbiome. Iam assuming BL1, 2 3, 4 are the insect samples, but perhaps a compariosn of the gut micorbiomes together with the different diets will show that despite the varied manures, the larvae selective screen for gut microbe. This rationale an be incorprotaed into the introduction nd the discussion

Comments on the Quality of English Language

The revised submission is signfcanlty more improved, but the introduction, results and discussion could use some further restructuring and a slght shift in focus given the data generated to make it more impressive and advance knoweldge.

1.    I appreciate the revisions made by the authors. The mansucript is much more coherent in this version. However, the introduction could use some further background. The mention of diet is a good adition, but the switch to manure has no hsitorical citations provided. Do we know if the different manusre result in different gut micprbiobiota in other insect or other sarcopjagous insects? If not, then framw the intro as the need to investigate this given the possible utility of this insect for forensic or nutritional purposes.

2.     I appreciate the inclusion of the diets in the results but fail to see the comparison (atleast in an NMDS plot and mayeb even an alpha diversity comparison) of the communities in the diet and how they differ from the insects. This would indicate some selective screening lending credenc to the presence of a “core stabel “ gut microbiome. Iam assuming BL1, 2 3, 4 are the insect samples, but perhaps a compariosn of the gut micorbiomes together with the different diets will show that despite the varied manures, the larvae selective screen for gut microbe. This rationale an be incorprotaed into the introduction nd the discussion

Author Response

Dear Reviewer:

Thank you for dedicating your time to reviewing this manuscript. Please find below the detailed responses and the corresponding revisions/corrections, which have been highlighted as track changes in the re-submitted files.

Regards

Guanjie
